# Comparative efficacy and tolerability of new-generation antidepressants for major depressive disorder in children and adolescents: protocol of an individual patient data meta-analysis

Xinyu Zhou,[1] Andrea Cipriani,[2,3] Toshi A Furukawa,[4] Pim Cuijpers,[5] Yuqing Zhang,[6,7] Sarah E Hetrick,[8] Juncai Pu,[6,7] Shuai Yuan,[6,7] Cinzia Del Giovane,[9] Peng Xie[6,7]

## ABSTRACT

**Introduction** Although previous conventional meta-analyses and network meta-analyses have provided some important findings about pharmacological treatments for children and adolescents with depressive disorders in the past decades, several questions still remain unsolved by the aggregate data from those meta-analyses. Individual participant data meta-analysis (IPD-MA) enables exploration of the impacts of individual characteristics on treatment effects, allowing matching of treatments to specific subgroups of patients. We will perform an IPD-MA to assess the efficacy and tolerability of new-generation antidepressants for major depressive disorder in children and adolescents.

**Methods and analysis** We will systematically search for all double-blind randomised controlled trials (RCTs) that have compared any new-generation antidepressant with placebo for the acute treatment of major depressive disorder in children and adolescents, in the following databases: PubMed, EMBASE, the Cochrane Library, PsycINFO, Web of Science, CINAHL, LILACS and ProQuest Dissertations. We will contact all corresponding authors of included RCTs and ask for their cooperation in this project by providing individual participant data from the original trials. The primary outcomes will include efficacy, measured as the mean change of depression symptoms by Children's Depression Rating Scale Revised (CDRS-R), and tolerability, measured as the proportion of patients who withdrew from the trials early due to adverse effects. The secondary outcomes will include response rates, remission rates, deterioration rate, all-cause discontinuation, suicidal-related outcomes and global functioning outcome. Using the raw de-identified study data, we will use mixed-effects logistic and linear regression models to perform the IPD-MAs. The risk of bias of included studies will be assessed using the Cochrane risk of bias tool. We will also detect the publication bias and effects of non-participation of eligible studies.

**Dissemination** Ethical approval is not required given that informed consent has already been obtained from the patients by the trial investigators before the included trials were conducted. This study may have considerable implications for practice and help improve patient care.

**PROSPERO registration number** CRD42016051657.

### Strengths and limitations of this study

► The study will use individual patient data that can take into account within-study and between-study differences and yield more reliable estimates of treatment effects than meta-analysis of aggregate data.

► Individual patient data meta-analysis can provide insight into the patient groups most likely to benefit from new-generation antidepressants and the most effective kinds of antidepressants.

► It is difficult to ensure all trials were identified because not all trials are registered, especially for these old trials.

► The another difficulty of this study will be collecting the patient-level information from all eligible trials, for some of the original investigators may not be willing or able to share the data. For example, for the fluoxetine trials, European Medicines Agency did not have them and Medicines and Healthcare Products Regulatory Agency were to have saved the records but they could only find three placebo controlled double blind randomised controlled trials; the other records had been destroyed as per their policy of older reports.

► We found that different clinical study report (CSRs) depending on the company and the time of the study varied significantly with respect to quality. Therefore, this definitely data would rely on getting access to databases, among others, for complete data.

## BACKGROUND

Major depressive disorder (MDD) is a commonly occurring serious mental disorder, accounting for a large portion of the global burden of disease. The overall prevalence rate of depressive disorder is about 3% in children and 6% in adolescents.[1] Depressive disorder in youth is often associated with high rates of comorbid mental disorders, functional impairment and suicide.[2–5] For young

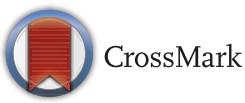

For numbered affiliations see end of article.

**Correspondence to**
Professor Peng Xie;
xiepeng973@126.com

people aged 10–19 years, depressive disorders are the leading cause of health-related burden, accounting for 6%–10% of the disability-adjusted life-years.[6] Early-onset depression is an important predictor of the recurrence of depressive disorders. In a naturalistic follow-up study, up to 55% paediatric patients who recovered from the first episode of MDD had a second episode within 5 years and rose to 72% within 15 years.[7]

In the past 20 years, several new-generation antidepressants have been found to be effective in the treatment of adult MDD.[8 9] However, whether to use antidepressants in children and adolescents are still matters of controversy, mainly due to concerns about efficacy and potentially increased risk of treatment-emergent suicide in those young patients.[10 11] In 2004, some worrying interpretations from a conventional meta-analysis were shown: published data suggested a favourable risk benefit profile for some selective serotonin reuptake inhibitors (SSRIs); however, addition of unpublished data indicated that risks could outweigh the benefits of these drugs (except fluoxetine) for the treatment of depression in children and young people.[12] Recently, our published network meta-analysis showed that most currently available antidepressants do not seem to offer a clear advantage over placebo for depression in children and adolescents, and fluoxetine is probably the best option to consider when a pharmacological treatment is indicated.[13] Nevertheless, several questions still remain unsolved by the aggregate data from conventional and network meta-analyses. First, the effect sizes of some antidepressants in previous meta-analyses had large confidence/credible interval with its upper limit close to the point of no difference, which raises the question of whether this estimate is robust enough to inform clinical practice.[14] Second, most studies included both children and adolescents, but they did not separately report the data of different age groups. Thus, it remains unclear whether the antidepressants are efficacious across the diverse populations included. Third, there was a strict range of baseline severity scores included in these previous meta-analyses. For example, in our previous network meta-analysis (NMA) analysis, most studies focused on samples with moderate to severe depressive severity, with few trials of those with mild to moderate or very severe range. Therefore, whether the antidepressants have similar efficacy for mildly or severely depressed patients is another important question that remains. Fourth, RCTs evaluating antidepressant treatments in children and adolescents seldom report the number of patients who deteriorated during treatment; thus, it is not possible to investigate mean deterioration effects found in randomised trials and its moderators using conventional and network meta-analytical approaches.

Individual participant data meta-analysis (IPD-MA) is an increasingly popular approach for synthesising and investigating treatment effect estimates. IPD-MA has many statistical and clinical advantages over meta-analyses of aggregate data. For example, clinical heterogeneity can be reduced by controlling for patient-level covariates in IPD-MA,[15 16] which offers the potential to explore additional, more thorough and potentially more appropriate analyses compared with those possible with aggregate data.[17] IPD-MA also provides unique opportunities to identify underlying individual characteristics as prognostic factors or negative effects across several studies.[18] Therefore, we will perform an IPD-MA to assess the efficacy and tolerability of new-generation antidepressants for MDD in children and adolescents.

## METHODS
### Criteria for included studies
#### Types of studies
Studies included in this IPD-MA will be double-blind randomised controlled trials (RCTs), including studies with cluster or cross-over designs. Given possible carry-over effects, we will only consider data from the first study period in cross-over trials. We will exclude trials employing inappropriate randomisation strategies, such as quasi-randomised designs.

#### Types of participants
Studies will be included in the IPD-MA if they aim at (1) children and adolescents aged between 6 and 18 years when initially enrolled in the studies and (2) with primary diagnosis of MDD according to standard diagnostic criterion, such as the Diagnostic and Statistical Manual of Mental Disorders[19–23] or the International Classification of Diseases.[24 25] Studies will be excluded if they included patients with bipolar depression or treatment-resistant depression, while patients with comorbid general psychiatric disorders, such as anxiety disorder, will not be excluded.

#### Types of interventions
We will include all RCTs comparing any new-generation antidepressant with placebo during the acute treatment phase of depression in children and adolescents. The following new-generation antidepressants using prescribed oral and therapeutic dose range will be included.[8 13 26]

1. SSRIs, for example, fluoxetine, fluvoxamine, paroxetine, sertraline, citalopram and escitalopram.
2. Serotonin and norepinephrine reuptake inhibitors (SNRIs), for example, duloxetine, venlafaxine, desvenlafaxine, milnacipran and levomilnacipran.
3. Other antidepressants, for example, mirtazapine, mianserin, nefazodone, trazodone, vortioxetine, vilazodone, bupropion, reboxetine and agomelatine.

We will only include RCTs with a minimum of 4-week treatment duration because the onset of benefit for most antidepressants often takes at least 4 weeks.[27] We will exclude trials designed as maintenance treatment or relapse prevention, unless outcome data from the acute phase can be accessed separately. Combination studies

and augmentation studies (eg, combined with different antidepressant or psychotherapy) will also be excluded.

## Types of outcome measures
### Primary outcomes
Overall efficacy: The primary outcome of efficacy will be the overall change in depressive symptoms, as measured using Children's Depression Rating Scale Revised (CDRS-R)[28] from baseline to endpoint. For RCTs that didn't measure CDRS-R, we will try to convert other depression scales (such as (Hamilton depression scale) HAMD[29] or (Montgomery–Åsberg Depression Rating Scale) MADRS)[30] scores to CDRS-R scores, by using a factor derived from the RCTs that used both scales.

As shown in our previous network meta-analysis,[13] trial duration varied from 6 weeks to 36 weeks, and the majority of trials employed a treatment duration of 8 weeks. We will try to obtain repeated measures from individual trials if possible. To improve comparability between the included trials, we will prefer the data from 8-week (or the closest to 8 week) time point for efficacy outcomes.

Overall tolerability: The tolerability of treatment will be the proportion of patients who drop out of the trials early due to side effects at the end of the blinded treatment.

### Secondary outcomes
Response rate: Response rate will be defined as 50% reduction from baseline to endpoint on CDRS-R (or another standardised rating scale such as HAMD or MADRS).

Remission rate: Remission rate will be defined as the CDRS scores of less than 28.[31]

Deterioration rate: Deterioration represents the depression symptom severity increases after treatment. Deterioration rate will be defined as the proportion of patients whose CDRS-R scores from baseline to endpoint had reliable change index below the cut-off of −1.96.[32]

Overall acceptability: The acceptability of treatment will be the proportion of patients who drop out of the trials early for any cause at the end of the blinded treatment.

Suicide-related outcomes: Suicide-related dichotomous and continuous outcomes will be measured. We will extract the number of participants with suicide-related events (combined suicidal ideation and suicidal behaviour) during the acute treatment, as measured on a standardised, validated and reliable rating scale or reported cases of suicidal ideation and behaviour.[33] In addition, if data are available, we will also collect data on suicidal ideation as a continuous outcome where a standardised, validated and reliable rating scale, such as the Suicidal Ideation Questionnaire-Junior High School version,[34] has been used.

Global functioning: The outcome of global functioning will be the overall change in validated scales from baseline to endpoint. The commonly used tools of functioning scales included the Children's Global Assessment Scale,[35] Global Assessment of Functioning[36] and so on.

Aggressive behaviour: The outcome of aggressive behaviour will be the proportion of cases who reported the aggressive behaviour, such as hostility and assault, during the acute treatment.[37 38]

## Data sources and search strategy
We will first include the RCTs identified by the criteria used in our previous work,[13 39] and then we will update the extensive searching to bring it up to date. Briefly, we will identify any published and unpublished RCTs, in any language, from electronic systematic searches of PubMed, EMBASE, the Cochrane Library, PsycINFO, Web of Science, LILACS, CINAHL and ProQuest Dissertations. Electronic databases will be searched with free words and Medical Subject Headings terms using the following strategy: (depress* or dysthymi* or mood disorder* or affective disorder*), combined with (adolesc* or child* or boy* or girl* or juvenil* or minors or paediatri* or pediatri* or pubescen* or school* or student* or teen* or young or youth*) and combined with a list of antidepressants, including (selective serotonin reuptake inhibitors or SSRI or fluoxetine or fluvoxamine or paroxetine or sertraline or citalopram or escitalopram or serotonin and norepinephrine reuptake inhibitors or SNRI or duloxetine or venlafaxine or desvenlafaxine or milnacipran or levomilnacipran or mirtazapine or mianserin or nefazodone or trazodone or vortioxetine or vilazodone or bupropion or reboxetine or agomelatine). In addition, we will also identify additional trials and unpublished data by searching: (1) international trials registries, mainly including of ClinicalTrials.gov and WHO International Clinical Trials Registry Platform; (2) US Food and Drug Administration (FDA) reports; (3) the International Prospective Register of Systematic Reviews (PROSPERO); (4) websites of main manufactures, for example, GlaxoSmithKline, Lilly, Organon, Forest Pharmaceuticals and Bristol-Myers Squibb; (5) manual hand-search of key journals and conference proceedings, for example *J Child Adolesc Psychopharmacol, J Am Acad Child Adolesc Psychiatry, Child Adolesc Psychiatry Ment Health, Psychopharmacol Bull, Arch Gen Psychiatry, Am J Psychiatry, Eur Psychiatry* and *Depress Anxiety*. Additional relevant RCTs will be obtained by hand-searching reference lists of included studies and relevant reviews. We will also contact corresponding authors of included RCTs, manufactures, FDA and other possible institutions for unpublished trials.

## Study selection and data extraction
### Selection of trials
We will first manually remove duplicates of initial search results, and then two experienced reviewers will independently screen titles and abstracts from the retrieved results for possible candidates. We will exclude the trials in which both reviewers judge they do not meet eligibility criteria. Full texts of all remaining papers will be retrieved, and two reviewers will independently examine whether to include them by the same eligibility criteria. Any difference of opinion, for each step, between the reviewers will be resolved through discussion with another member of the reviewing team or by contacting the authors of the

trials for clarification. The selection process of retrieved studies and the reasons for exclusion of trials (eg, ineligible populations, not randomised trials) will be shown in a flow chart.

### Data collection

From the included RCTs, two reviewers will independently extract the trial level information using standardised data collection forms, including trial characteristics, patient characteristics, intervention details and any other information relevant to this review.

We will contact all corresponding authors or sponsor pharmaceutical companies of included RCTs and ask for their cooperation in this project. The corresponding authors' contact information will be abstracted from the papers, online research profiles (eg, Google Scholar) or other available ways. Specifically, we will (1) send emails to the authors explaining the study purpose and invite them to cooperate in this project; (2) send reminder emails 4 and 6 weeks later if no response; and (3) contact the corresponding authors by phone or possible personal contacts. We will also report on the process of interaction with the sponsor companies, as applicable.

Trial-level information and individual participant data to be obtained from the original authors are shown in table 1, respectively. The raw data can be provided in any convenient manner (such as by email) in common types of electronic format, such as Excel, SPSS, Stata and so on . All obtained data will be converted to a uniform format and saved on a secure server at the Chongqing Medical University. The data set will not contain any personal identifier of patients, such as names or phone numbers. Only authorised members of the research team will be allowed to access the data set.

### Data checking

We will check for data-entry mistakes and consistency and reanalyse the data within each study according to the original statistical methodology; the results will be compared with the published summary results. Any error will be resolved by discussion with the original investigators, and data corrections will be made if necessary.

### Missing data

Handling of missing data will depend on the proportion of missing data in the full data set. In general, we will prefer to manage missing data for both patient characteristics and outcomes through multiple imputation (MI) methods, such as MI and mixed-effects model repeated measures (MMRM), because MI techniques with a missing at random assumption tends to yield more unbiased results than single imputation methods.[40] Missing data will be imputed using the command *mi impute mvn* in Stata V.14.0. However, if we obtain repeated measures from individual trials, we will use MMRM approach.

### Risk of bias assessment and quality of study

Two independent review authors will use the Cochrane Collaboration's risk of bias' tool[41] to evaluate the methodological and hence bias risk of eligible studies, and quality assessment will be reported on a study level. The risk of bias will be assessed across seven items, including random sequence generation, allocation concealment, blinding of intervention, blinding of outcome assessment, incomplete outcome data, selective outcome reporting and other bias (eg, conflicts of interests) with three levels of risk (high, unclear, low). We will rate the quality of study as follows: high-risk study (two or more items rated as high risk of bias); low-risk study (five or more items rated as low risk and no more than one as high risk); unclear risk study (all remaining situations). Any disagreements will be resolved by consensus or consulting the original authors.

### Publication bias and effects of non-participation of eligible studies

We will use contour enhanced funnel plot to detect publication bias for study level data (full set of studies meeting inclusion criteria) and patient-level data (the set of studies that were included in the IPD-MA), if at least 10 studies are available.[42] We will also use Egger's test to quantify the bias, with a P value <0.10 taken to indicate statistical evidence of asymmetry.[43] In order to examine the effects of non-participation of eligible studies, we will conduct a meta-regression analysis with the effect size of primary outcomes (based on study level data) as the dependent variables and whether or not the patient-level data are included as the predictor indicating. The analyses will be conducted in Stata V.14.0.

### Statistical analysis

All analyses will be performed by intention-to-treat analysis. Descriptive statistics will be presented as mean (SD) or median (IQR) for continuous variables and number (per cent) for categorical variables.

#### Individual patient data meta-analyses

We will first use the one-stage approach to perform the IPD-MAs, as it offers the highest degree of flexibility for making necessary assumptions[44] and uses a more exact statistical approach than two-stage approach.[45] We will perform analyses in Stata with the commands *mixed* (for linear random-effects models), *meqrlogit* (for logistic models) and *ipdforest* (for forest plot).[46] To account for between study differences, we will use mixed-effects logistic models for categorical outcomes and mixed-effects linear regression models for continuous outcomes. Treatment assignment will be introduced as a fixed-effects variable 'treatment'. As outcomes might vary across studies, we will force the 'study' and the interaction term 'study*treatment' as random-effects variables into all models. The important clinical and demographic predictors variables (eg, sex,[47] age,[48] baseline severity score[49] and treatment duration) will be used as regressors in the models. The heterogeneity of treatment effects across studies will be assessed using the $I^2$ statistic.[50] Finally, we will carry out the following sensitivity analyses of the primary outcomes:

**Table 1** Data items to be requested for individual participant data meta-analysis

| Trial-level information | Demographic and baseline characteristics | Therapeutic process | Outcomes |
|---|---|---|---|
| 1. Study protocol<br>2. Clinical study report (if available);<br>3. List of publications<br>4. Setting (such as primary care, hospitals, clinics)<br>5. Information about the risk of bias (sequence generation, allocation of concealment, masking and ITT analysis)<br>6. Any other information relevant to this review | 1. Unique identification number for anonymity<br>2. Date of randomisation<br>3. Sex (male, female)<br>4. Race (White/Caucasian, African/African-American, Asian, multiracial, other)<br>5. Body mass index, kg/m$^2$<br>6. Height, cm<br>7. Weight, kg<br>8. Age, year<br>9. Age at onset, year<br>10. Length of illness, month<br>11. Number of MDD episodes<br>12. Duration of current episode, month<br>13. Baseline depression symptom score<br>14. Baseline quality of life and functioning score<br>15. Previous and/or ongoing secondary psychiatric disorder (anxiety disorder, externalising disorder (ADHD, conduct disorder, oppositional defiant disorder), psychotic disorder, substance-related disorder)<br>16. Family history of MDD<br>17. Household (two parents, other)<br>18. Number of siblings<br>19. Life and social history<br>20. Previous suicide attempt | 1. Treatment (antidepressant, placebo)<br>2. Dose range<br>3. Total actual drug exposure<br>4. Treatment duration<br>5. Co-prescriptions other than antidepressant<br>6. Prior treatments (no therapy, psychotherapy, pharmacotherapy, both psychotherapy and pharmacotherapy) | 1. Depression symptom scores at each evaluation (scale, time point)<br>2. Quality of life and functioning scores at each evaluation (scale, time point)<br>3. Study discontinuation and reason (drop out before starting the treatment, lack of efficacy, adverse events, others)<br>4. Adverse events<br>5. Serious adverse events<br>6. Suicide-related event |

ADHD, attention deficit hyperactivity disorder; ITT, intention to treat; MDD, major depressive disorder.

(1) excluding trials with a follow-up longer than 12 weeks and (2) excluding studies where HAMD and MADRS scores were mapped onto CDRS-R.

## Ethics and dissemination

This protocol is registered in PROSPERO at the National Health Service Centre for Reviews and Dissemination at the University of York (registration number: CRD42016051657). No ethics review is required for this IPD-MA, since informed consent has already been obtained from the patients by the trial investigators before the trial was conducted. We will publish the results in a peer-reviewed journal.

**Author affiliations**
¹Department of Psychiatry, The First Affiliated Hospital of Chongqing Medical University, Chongqing, China
²Department of Psychiatry, University of Oxford, Oxford, UK
³Oxford Health NHS Foundation Trust, Warneford Hospital, Oxford, UK
⁴Department of Health Promotion and Human Behavior, Kyoto University Graduate School of Medicine and School of Public Health, Kyoto, Japan
⁵Department of Clinical Psychology, Neuro and Developmental Psychology, Amsterdam Public Health Research Institute, VU University Amsterdam, Amsterdam, The Netherlands
⁶Department of Neurology, The First Affiliated Hospital of Chongqing Medical University, Chongqing, China
⁷Institute of Neuroscience and the Collaborative Innovation Center for Brain Science, Chongqing Medical University, Chongqing, China
⁸Orygen, The National Centre of Excellence in Youth Mental Health and the Centre of Youth Mental Health, University of Melbourne, Melbourne, Australia
⁹Institute of Primary Health Care (BIHAM), University of Bern, Bern, Switzerland

**Acknowledgements** AC is supported by the National Institute for Health Research Oxford Cognitive Health Clinical Research Facility.

**Contributors** PX and XZ conceived the study and wrote the first draft of the protocol and will assist with the data extraction and analysis and draft the results and discussion sections. AC, TAF, PC, SEH and CDG assisted in protocol design and revision and will help draft the final manuscript. XZ, YZ, JP and SY participated in the search strategy development and will carry out most of the data collection. CDG, AC and TAF participated in the design of data synthesis and analysis and will carry out the statistical analyses. All the authors have approved the publication of the protocol.

**Funding** This work is supported by the National Basic Research Program of China (973 Program) (Grant No. 2009CB918300).

**Disclaimer** The funders had no role in the protocol design; the writing of the protocol; or the decision to submit the protocol for publication.

**Competing interests** AC reports personal fees from Accord Healthcare as an expert witness for a patent issue about quetiapine extended release. TAF has received lecture fees from Eli Lilly, Janssen, Meiji, MSD, Otsuka, Pfizer and Tanabe-Mitsubishi and consultancy fees from Sekisui Chemicals and Takeda Science Foundation. He has received royalties from Igaku-Shoin and Nihon Bunka Kagaku-sha publishers. He has received research support from Mochida and Tanabe-Mitsubishi. SEH is an editor of the Cochrane Common Mental Disorders Group, an author of the Cochrane systematic review of newer generation antidepressants for depression in children and adolescents and an author (senior) on the Cochrane review of psychological, pharmacological and their combination for child/adolescent depression. XZ, PC, YZ, JP, SY, CDG and PX declare no competing interests.

**Provenance and peer review** Not commissioned; externally peer-reviewed.

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
