## [Reviewer comments · BMJ Open]

ARTICLE DETAILS

TITLE (PROVISIONAL)	Comparative efficacy and tolerability of new-generation antidepressants for major depressive disorder in children and adolescents: protocol of an individual patient data meta-analysis
AUTHORS	Zhou, Xinyu; Cipriani, Andrea; Furukawa, Toshi; Cuijpers, Pim; Zhang, Yuqing; Hetrick, Sarah; Pu, Juncai; Yuan, Shuai; Del Giovane, Cinzia; Xie, Peng

VERSION 1 – REVIEW

REVIEWER	Sverre Bergh Innlandet hospital trust, Norway
REVIEW RETURNED	01-Sep-2017

GENERAL COMMENTS	Thank you for submitting the protocol of an important study that you are about to perform. It is important for the patients, their relatives and the health-care system to be able to answer whether new-generations antidepressants are useful for children and youngsters with MDD. The IPD-MA will add useful information to previous meta-analysis, and by improving power be of better quality than previous single RCT. I do only have a few comments to the authors: 1. Would it be wise to search PROSPERO as well for other systematic reviews? To find studies not found in the other databases?2. I am puzzled by the exclusion of studies including patients with treatment resistant depression. Why would that be? Any chance of biasing results if they are excluded?
--

REVIEWER	Tarang Sharma Nordic Cochrane Centre, Denmark None relevant but would like to state that I am currently working on issues around publication bias and selective reporting within antidepressant trials.
REVIEW RETURNED	12-Sep-2017

GENERAL COMMENTS	1. General comments:• A very interesting protocol for a potentially good systematic review using IPD data (depending on availability), which has not been done before in this area so this is a positive move. Getting an understanding of which type of patients are particularly harmed, in terms of increased suicidal ideation and behaviour would be definitely a step in the right direction.
---

• Though the authors are making efforts to identify all possible unpublished trials, getting access to this data is a big risk to the project.

Due to lawsuits, the full CSRs for GSK paroxetine for children and adolescents (except Case Report Forms: CRFs) are available with IPD data through the appendices. <https://www.gsk-clinicalstudyregister.com>

As seen by the case of study 329 which showed that CRFs were only available on request (and only in a desktop online version) and that they showed furthermore that even the IPD data from the tables had some suicidal ideation events missing (LeNoury et al. 2015, BMJ) – so some data will even be missed without CRFs.

I would encourage the authors to at least consider the CSRs and ideally CRFs for paroxetine that are available publicly, as their results would otherwise be misleading and not reliable. This way they can estimate what the difference (as the value differences between the drugs of the same class have been shown to be minimal, except fluoxetine) would be for the estimates if they had such data for all the drugs.

Ideally, but of course depending on resources they can request all data from the participating companies using: <https://clinicalstudydatarequest.com>

• Missing aggression: as increased aggression has been shown to be more than doubled within this population (Sharma et al. 2016, BMJ), and has been identified as increased for many years through both the work of FDA (2006) and the MHRA (2004) it is a mistake and a missed opportunity to not include this outcome, especially as the potential for understanding which persons are most at risk using IPD data. This is especially of interest with respect to high level of high shootings etc.

• It is a review, which has been registered in PROSPERO and the authors are also submitting it here so that its publicly available (and then becomes MEDLINE searchable) is definitely a plus.

2. Strengths and limitations of the study

• I would recommend that the limitations (last bullet) only discuss availability of data would separate that into different points as per the strengths.

- Firstly issue of ensuring they have identified all trials (not all trials are registered, this is especially the case for old trials).

- Then secondly, getting access to data: not all will be available or authors willing to give the data. For example for the fluoxetine trials, EMA did not have them and MHRA were to have saved the records but they could only find three placebo controlled double blind RCTs, the other records had been destroyed as per their policy of older reports.

- Then thirdly, and finally the issue of quality; we found that different CSRs depending on the company and the time of the study varied significantly with respect to quality, therefore data. So this definitely would rely on getting access to databases etc for complete data.

3. Criteria for included studies

• Types of studies – It is encouraging to see that only the first study period for cross-over trials will be used.

	• Types of interventions – excluding trials less than 4 weeks is great, maybe sensitivity analyses with trials of 4-8 weeks and 8 weeks or longer could be interesting to see for the primary outcomes. 4. Outcome measures • Very encouraging to see that attempts of mapping HAMD and MADRS scores onto CDRS-R will be made such that more data can be seen on the same scale perhaps a sensitivity analysis with and without mapped data could be shown. • Seeing the outcomes of tolerability and acceptability that are very good hard outcomes, not solely based on subjective rating scales are very welcome.• Additionally, seeing the outcome for deterioration was a very interesting one I having seen previously in this area and a good addition.• Inclusion of an overall functioning scale is also very valuable.• Please note the general comment on missing aggression. 5. Data sources and search strategies • Please refer to the general and limitation comments on publication bias and selective reporting. 6. Data collection • I would recommend the authors to report on the process of interaction with the companies. Many have agreed on the principle of “All Trials” and sharing data but in our work, we found that in practice this wasn’t always the case. 7. Missing data • The methods used for imputing missing data are very thorough and should help minimise erroneous results and therefore interpretations. 8. Risk of bias • In our work and work done by Tom Jefferson, and Peter Doshi and colleagues on Tamiflu found that the standard RoB is maybe insufficient when looking at CSRs. I would recommend considering at least the conflicts of interests within the other bias criteria (which will be introduced as a required separate domain with the next iteration of the RoB tool).
--	---

VERSION 1 – AUTHOR RESPONSE

Reviewer: 1

Reviewer Name: Sverre Bergh

Institution and Country: Innlandet hospital trust, Norway

Please state any competing interests or state 'None declared': Noen declared

Please leave your comments for the authors below

Thank you for submitting the protocol of an important study that you are about to perform. It is important for the patients, their relatives and the health-care system to be able to answer whether new-generations antidepressants are useful for children and youngsters with MDD.

The IPD-MA will add useful information to previous meta-analysis, and by improving power be of better quality than previous single RCT.

I do only have a few comments to the authors:

1. Would it be wise to search PROSPERO as well for other systematic reviews? To find studies not found in the other databases?

Authors' response

Thanks for your suggestion, we have added PROSPERO to the list of databases to search.

2. I am puzzled by the exclusion of studies including patients with treatment resistant depression. Why would that be? Any chance of biasing results if they are excluded?

Authors' response

We can see the point raised by Dr Bergh and agree that the exclusion of patients treatment resistant depression may limit the applicability of the results to these clinical subgroups, however this decision is intended as a methodological strength to assure transitivity in the network. Patients with treatment resistant depression are usually prescribed second line treatments or combination of two or more different interventions (including antipsychotic, lithium, mood stabilisers and non-pharmacological strategies such as rTMS). If we included these patients, it would be a clear violation of the transitivity assumption (Cipriani et al., Lancet 2016).

Reviewer: 2

Reviewer Name: Tarang Sharma

Institution and Country: Nordic Cochrane Centre, Denmark

Please state any competing interests or state 'None declared': None relevant but would like to state that I am currently working on issues around publication bias and selective reporting within antidepressant trials.

Reviewer: Tarang Sharma, Research and PhD Fellow, Nordic Cochrane Centre;

University of Copenhagen.

Peer review for: Comparative efficacy and tolerability of new-generation antidepressants for major depressive disorder in children and adolescents: protocol of an individual patient data meta-analysis.

1. General comments:

- A very interesting protocol for a potentially good systematic review using IPD data (depending on availability), which has not been done before in this area so this is a positive move. Getting an understanding of which type of patients are particularly harmed, in terms of increased suicidal ideation and behaviour would be definitely a step in the right direction.

Authors' response

Thank you (no comment needed)

- Though the authors are making efforts to identify all possible unpublished trials, getting access to this data is a big risk to the project. Due to lawsuits, the full CSRs for GSK paroxetine for children and adolescents (except Case Report Forms: CRFs) are available with IPD data through the appendices. <https://www.gsk-clinicalstudyregister.com>. As seen by the case of study 329 which showed that CRFs were only available on request (and only in a desktop online version) and that they showed furthermore that even the IPD data from the tables had some suicidal ideation events missing (LeNoury et al. 2015, BMJ) – so some data will even be missed without CRFs. I would encourage the authors to at least consider the CSRs and ideally CRFs for paroxetine that are available publically, as their results would otherwise be misleading and not reliable. This way they can estimate what the difference (as the value differences between the drugs of the same class have been shown to be minimal, except fluoxetine) would be for the estimates if they had such data for all the drugs. Ideally, but of course depending on resources they can request all data from the participating companies using: <https://clinicalstudydatarequest.com>

Authors' response

Thank you for your suggestions, we will use all available data in any format and from any available source and this is exactly what we planned to do when we reported in the protocol our search strategy (page 9): “..(4) websites of main manufactures, e.g., GlaxoSmithKline, Lilly, Organon, Forest Pharmaceuticals, Bristol-Myers Squibb;”

- Missing aggression: as increased aggression has been shown to be more than doubled within this population (Sharma et al. 2016, BMJ), and has been identified as increased for many years through both the work of FDA (2006) and the MHRA (2004) it is a mistake and a missed opportunity to not include this outcome, especially as the potential for understanding which persons are most at risk using IPD data. This is especially of interest with respect to high level of high shootings etc.

Authors' response

Thank you for your suggestion, we have added now this outcome and the corresponding references

- It is a review, which has been registered in PROSPERO and the authors are also submitting it here so that its publically available (and then becomes MEDLINE searchable) is definitely a plus.

Authors' response

Thank you (no comment needed)

2. Strengths and limitations of the study

- I would recommend that the limitations (last bullet) only discuss availability of data would separate that into different points as per the strengths.
- Firstly issue of ensuring they have identified all trials (not all trials are registered, this is especially the case for old trials).

- Then secondly, getting access to data: not all will be available or authors willing to give the data. For example for the fluoxetine trials, EMA did not have them and MHRA were to have saved the records but they could only find three placebo controlled double blind RCTs, the other records had been destroyed as per their policy of older reports.
- Then thirdly, and finally the issue of quality; we found that different CSRs depending on the company and the time of the study varied significantly with respect to quality, therefore data. So this definitely would rely on getting access to databases etc for complete data.

Authors' response

We have added these limitations in the text, as suggested. Thank you.

3. Criteria for included studies

- Types of studies – It is encouraging to see that only the first study period for cross-over trials will be used.

Authors' response

Thank you (no comment needed)

- Types of interventions – excluding trials less than 4 weeks is great, maybe sensitivity analyses with trials of 4-8 weeks and 8 weeks or longer could be interesting to see for the primary outcomes.

Authors' response

This is an interesting point and we added a new statement, as follows (page 11): “Finally, we will carry out the following sensitivity analyses of the primary outcomes: (i) excluding trials with a follow up longer than 12 weeks ...”

4. Outcome measures

- Very encouraging to see that attempts of mapping HAMD and MADRS scores onto CDRS-R will be made such that more data can be seen on the same scale perhaps a sensitivity analysis with and without mapped data could be shown.

Authors' response

Thank you, on page 13 we added the following statement: “... and (ii) excluding studies where HAMD and MADRS scores were mapped onto CDRS-R.”

- Seeing the outcomes of tolerability and acceptability that are very good hard outcomes, not solely based on subjective rating scales are very welcome.
- Additionally, seeing the outcome for deterioration was a very interesting one I having seen previously in this area and a good addition.

Authors' response

Thank you (no comment needed)

- Inclusion of an overall functioning scale is also very valuable.

Authors' response

Thank you (no comment needed)

- Please note the general comment on missing aggression.

Authors' response

See our reply above about the same issue.

5. Data sources and search strategies

- Please refer to the general and limitation comments on publication bias and selective reporting.

Authors' response

See our replies above.

6. Data collection

- I would recommend the authors to report on the process of interaction with the companies. Many have agreed on the principle of "All Trials" and sharing data but in our work, we found that in practice this wasn't always the case.

Authors' response

We have added the sentence on Page 9: "We will also report on the process of interaction with the sponsor companies, as applicable."

7. Missing data

- The methods used for imputing missing data are very thorough and should help minimise erroneous results and therefore interpretations.

Authors' response

Thank you (no comment needed)

8. Risk of bias

- In our work and work done by Tom Jefferson, and Peter Doshi and colleagues on Tamiflu found that the standard RoB is maybe insufficient when looking at CSRs. I would recommend considering at least the conflicts of interests within the other bias criteria (which will be introduced as a required separate domain with the next iteration of the RoB tool).

Authors' response

Thanks for your suggestion, we have added the conflicts of interests within the "other bias" criteria (see page 10 of the revised manuscript).

VERSION 2 – REVIEW

REVIEWER	Dr. Sverre Bergh Innlandet Hospital trust, Norway
REVIEW RETURNED	15-Oct-2017

GENERAL COMMENTS	The Authors have responded adequately to the comments from the two reviewers in the first review round, either by adding more information to the manuscript or replying to the comment.
---

REVIEWER	Tarang Sharma Nordic Cochrane Centre and University of Copenhagen None. Currently researching issues around publication bias and selective reporting in antidepressant trials.
REVIEW RETURNED	23-Oct-2017

GENERAL COMMENTS	Two minor comments: 1. Study limitations: • Last point – is a copy paste from my comment, doesn't make sense for them to phrase it like that; consider editing. I would just state that different CSRs have different quality of data and therefore requesting access to the data within the databases/ raw data would be more beneficial for the authors. We found that different CSRs depending on the company and the time of the study varied significantly with respect to quality. Therefore, this definitely data would rely on getting access to databases etc for complete data. 2. Outcome measures Very happy to see the inclusion of aggression as a secondary outcome, it would have been a wasted opportunity if not included.
--